# Sustaining Student Roles, Digital Literacy, Learning Achievements, and Motivation in Online Learning Environments during the COVID-19 Pandemic

Zhonggen Yu 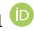

Department of English Studies, Faculty of Foreign Studies, Beijing Language and Culture University, Beijing 100083, China; yuzhonggen@blcu.edu.cn; Tel.: +86-189-5180-1880

**Abstract:** The sudden COVID-19 pandemic has forced many educational institutes to shut down and many students to stay home receiving online learning. This study aims to identify the changes in student roles and digital literacy and the strategies to improve motivation and learning achievement in online learning environments during the COVID-19 pandemic. Through a rapid evidence assessment review study based on the protocol of Preferred Reporting Items for Systematic Review and Meta-analysis (PRISMA), it is concluded that student roles have experienced great changes, that students have assumed multiple roles and that they are socially and cognitively engaged in their roles, roles that can be promoted by information technologies. Their digital literacy, individually varying, needs to be improved although it has undergone improvements. Digital technologies and social regulation can improve online learning achievements. Teaching strategies, teacher–student cooperation, gamification, and computer applications can improve online learning achievements. Future research could focus on inter-disciplinary research into the models to sustain online learning during or after the pandemic.

**Keywords:** student roles; learning achievements; digital literacy; motivation; online learning environments; COVID-19

## 1. Introduction

The sudden COVID-19 pandemic has forced many educational institutes to shut down and many students to stay home receiving online learning. In this isolated context, student roles have undergone dramatic changes. The students not only play the role of students, but also a role addressing familial issues, medical care, and even financial problems [1]. The online learning achievements have drawn the attention of educators, course designers, and students [2]. Those with lower digital literacy and weak motivation have possibly achieved less success in online learning environments. Researchers and teachers may thus take measures to improve teachers' and students' digital literacy and enhance their online learning motivational levels.

This study is timely since it aims to solve urgent problems during the COVID-19 pandemic. The lockdown environment has greatly influenced students' roles. Students have to receive online learning through online platforms where they receive less supervision but can share opinions more freely with peers through various digital tools such as social media and online platforms. This may exacerbate their learning achievements. In some underdeveloped areas, the digital infrastructures and the digital literacy of students are poor [3], which may dampen their online learning motivation, leading to poor learning achievements. It is thus necessary for educators to figure out measures to address these issues. This study will attempt to identify the changes in student roles and online learning achievements, followed by the ways to improve digital literacy and online learning motivation.



## 2. Literature Review

### 2.1. Student Roles

The outbreak of COVID-19 greatly challenges students who are forced to receive online education at home. They have reported that their support had been decreased and their student roles had been in conflict with other roles such as family members and breadwinners. Educators can make every effort to minimize these conflicts and improve their learning achievements. The pandemic has forced graduate students to adjust their new roles in remote or online learning approaches, where they learn in their student roles. However, inconsistent findings have been revealed. Students, as learning roles, positively evaluated the online learning methods and reported satisfaction with the virtual learning, which was proved to be an effective and feasible strategy to handle the emergency remote education [4]. Considering the inconsistent conclusions regarding student roles during the pandemic, we propose the research question: What are the changes in student roles during the COVID-19 pandemic?

### 2.2. Digital Literacy

Digital literacy, in this study, was operationally defined as the digital competencies in online, distance, or remote learning. Digital literacy was deemed as the multiple competencies regarding the application of digital technologies [5]. Digital literacy was concerned with students' capacity to adopt information technologies in online, offline, or blended contexts. Students' digital literacy was considered the essential factor influencing their engagement and motivation in online, offline, or blended learning [6]. Digital literacy is an indispensable element that can equip learners with a powerful learning capacity to receive education during or after the pandemic. Given the importance of digital literacy during the COVID-19 pandemic, we propose the research question: What are the changes in digital literacy during the COVID-19 pandemic?

### 2.3. Learning Achievements

The shift to online, remote, or distance learning has produced inconsistent findings in learning achievements during the COVID-19 pandemic. Online learning has also benefited those with disadvantaged economic and social statuses [7], who are able to stay at home receiving online learning with less expenditure. Although online education can address many difficulties caused by the lockdown, practical skills can hardly be trained or delivered through online platforms such as teleconferences and massive open online courses [8]. Online interactive skill training could be a challenge for online education [9]. For instance, clinical skills such as nursing or caring practice may be difficult to deliver through online methods [10]. Considering the inconsistent, even paradoxical findings, we propose the research question: How to improve learning achievements during the COVID-19 pandemic?

### 2.4. Motivation

Previous studies have revealed inconsistent findings regarding the levels of motivation in online learning environments during the COVID-19 pandemic. Learning motivation can exert a great influence on students' learning strategies, especially in pandemic-contextualized online education. However, participants tended to negatively assess online learning effectiveness, failing to be motivated to participate in online learning and address the various problems in the online environment [11]. A large number of learners negatively evaluated online education and held pessimistic attitudes towards the learning effectiveness. They neither believed they could achieve their learning goals through the online methods nor thought their communicative abilities could be enhanced through the online approach [12].

By contrast, students and teachers positively evaluated the impact of online learning on learning motivation. Before and after the online lecturing, teachers positively evaluated the motivation and knowledge acquisition. They opined that students became highly motivated, actively participatory, and more efficient when they learned knowledge through the

online approach [13]. Online English language education may enhance student motivation, self-efficacy, and meta-cognition in the isolated pandemic environment [14]. This may be because students encouraged themselves to engage in learning since their learning behaviors were not openly accessible to peers and teachers. Given that learning motivation may be difficult to improve in an online education, we propose the research question: How to improve online learning motivation during the COVID-19 pandemic?

## 3. Materials and Methods

This study first obtained sufficient literature by searching online databases based on their specific syntactic rules. The researchers then implemented a rapid evidence assessment review after adopting clustering techniques in VOSviewer, followed by the inclusion and exclusion of the literature based on the protocol of Preferred Reporting Items for Systematic Review and Meta-analysis (PRISMA) [15]. The PRISMA protocol has been widely used in systematic reviews [16–18]. The clustering could sort the research themes and the framework of PRISMA could help researchers select the highly influential literature, improving the quality of the review analysis.

Two experienced researchers searched several databases by keying in search terms according to the corresponding syntactic rules, ranging from their inception to 5 February 2022. The search terms included topic: ("online learning") AND topic: ("student role" OR "learning achievement *" OR "digital literacy" OR motivation OR "COVID-19" OR pandemic OR coronavirus). The online databases included Science Citation Index Expanded (SCI-EXPANDED) (from 1900 to 2022), Social Sciences Citation Index (SSCI) (from 1998 to 2022), Arts & Humanities Citation Index (A&HCI) (from 1998 to 2022), Conference Proceedings Citation Index—Science (CPCI—S) (from 1998 to 2022), Conference Proceedings Citation Index—Social Science & Humanities (CPCI—SSH) (from 1998 to 2022), Emerging Sources Citation Index (ESCI) (from 2017 to 2022), Current Chemical Reactions (CCR-EXPANDED) (from 1985 to 2022), and Index Chemicus (IC) (from 1993 to 2022). Both researchers obtained 3171 results. The research trend over two decades is shown in Figure 1.

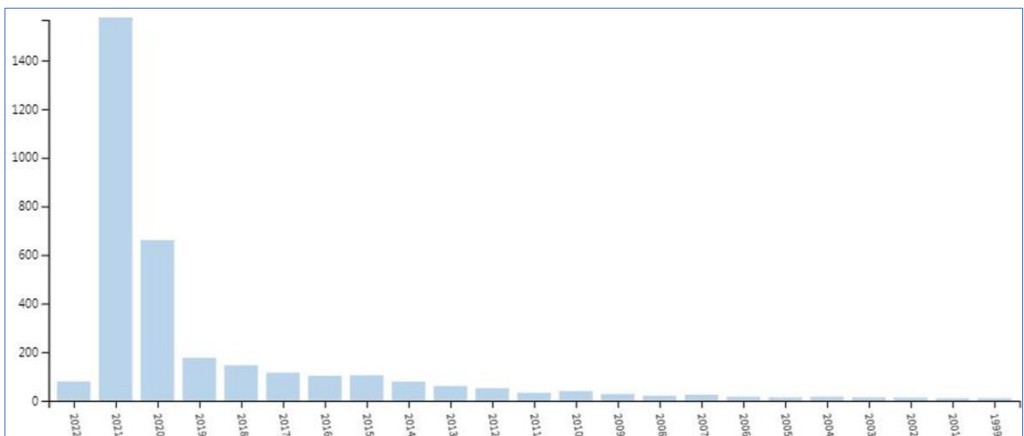

**Figure 1.** The research trend over two decades.

Figure 1 illustrates the publication trend over two decades in terms of student role, learning achievements, digital literacy, and motivation in online learning environments during the COVID-19 pandemic. The ordinate axis (y) in Figure 1 refers to the number of publications found in the databases, while the horizontal axis (x) indicates the publication years. Starting from the year 1999, researchers began to pay attention to online learning. The sparse publications fluctuated between 1999 and 2006. The years between 2007 and 2019 witnessed a gradual increase in online learning environments. The number of related publications rocketed up suddenly in the year 2020 with the outbreak of COVID-19, which continued to climb up until the peak in the year 2021. Accordingly, the pandemic has spurred the dramatic rise of publications regarding online learning. This trend also high-

lights the importance of student role, learning achievements, digital literacy, and motivation in online learning environments.

The researchers clustered the obtained results via VOSviewer to obtain the research themes. The selected keywords were classified into 11 clusters via VOSviewer (Figure 2). Cluster 1 included 83 items, e.g., academic achievement, academic emotions, academic motivation, achievement, and competence. Cluster 2 included 63 items, e.g., active learning, asynchronous learning, blended learning, cooperative learning, and student role. Cluster 3 included 60 items, e.g., adaptation, anxiety, burnout, digitalization, and COVID-19. Cluster 4 included 57 items, e.g., acceptance, instructor, interactivity, and self-efficacy. Cluster 5 included 55 items, e.g., augmented reality, game, and student performance. Cluster 6 included 52 items, e.g., collaboration, digital literacy, and peer feedback. Cluster 7 included 48 items, e.g., digital divide, facilitation, and interactive learning environment. Cluster 8 included 25 items, e.g., benefits and digital transformation. Cluster 9 included 15 items, e.g., cognitive presence and teaching presence. Cluster 10 included 3 items, i.e., badges, face-to-face learning, and gamification. Cluster 11 included 1 item, i.e., students' satisfaction.

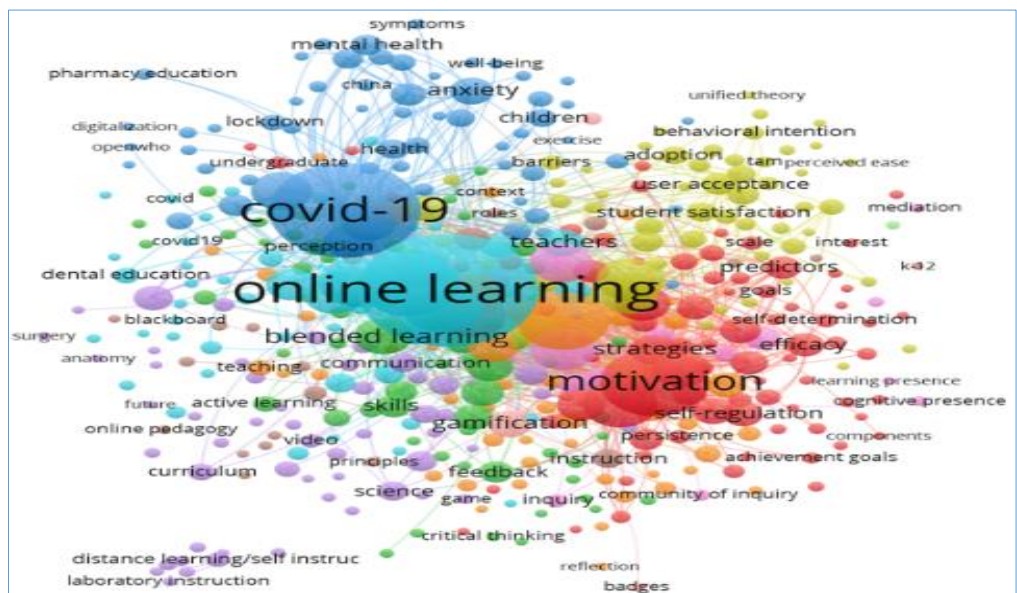

**Figure 2.** A clustering map of keywords.

In addition, VOSviewer provided the occurrences and total link strength for each selected keyword. The top frequently occurring keywords included online learning (occurrence = 1212, total link strength = 5228), COVID-19 (occurrence = 854, total link strength = 3468), motivation (occurrence = 440, total link strength = 2762), student role (occurrence = 321, total link strength = 2027), learning achievement (occurrence = 130, total link strength = 1000), digital literacy (occurrence = 37, total link strength = 140), performance (occurrence = 216, total link strength = 1551), satisfaction (occurrence = 178, total link strength = 1345), engagement (occurrence = 167, total link strength = 1119), self-efficacy (occurrence = 121, total link strength = 825), anxiety (occurrence = 74, total link strength = 435), and self-regulation (occurrence = 60, total link strength = 471). Considering occurrence, total link strength, and research theme, researchers focused their exploration on student role, digital literacy, online learning achievement, and online learning motivation.

Based on the flow of PRISMA, the two researchers finally included 57 studies (References 1–57) that guided this study (Figure 3). The literature would be excluded if they (1) were duplicate results, (2) were irrelevant to the research topic, (3) were out of the educational scope, (4) did not provide abstracts, (5) belonged to editorial collections or non-academic reports, (6) included small sample sizes with poor sampling strategies, (7) were not rigidly designed, (8) did not provide adequate information, or (9) arrived at unconvincing conclusions. The selection results of the two researchers reached high inter-rater

reliability (kappa coefficient $k$ = 0.91). In case both could not reach an agreement on any selection, a third examiner would be invited to determine it.

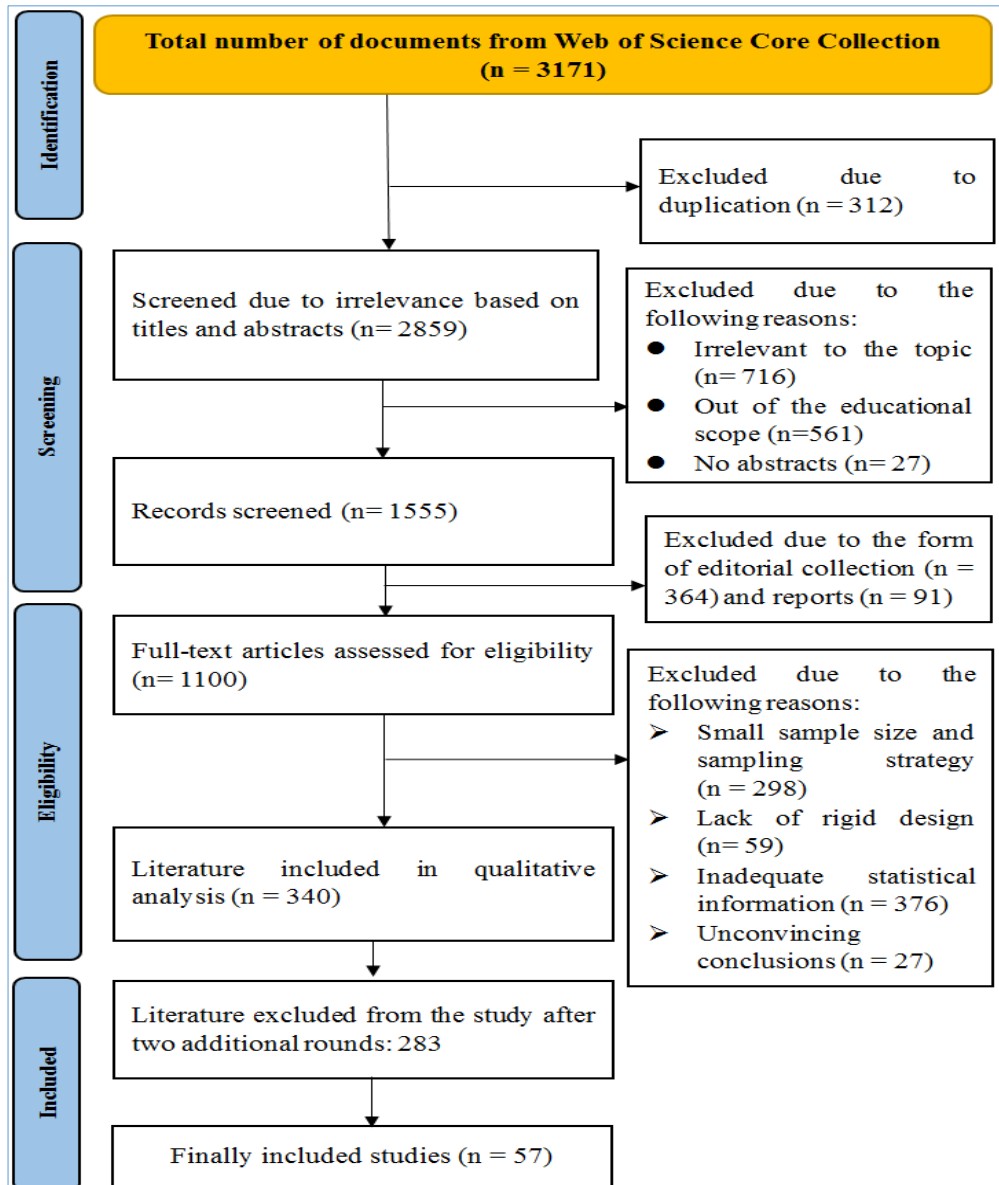

**Figure 3.** A PRISMA flow chart of literature inclusion.

## 4. Results

### 4.1. RQ1: What Are the Changes in Student Roles during the COVID-19 Pandemic?

Student roles have undergone great changes during the COVID-19 pandemic. Medical students have played an important role in containing the pandemic rather than learning. In the USA, medical students have played an essential role during the COVID-19 pandemic, where they formed an emergency response team to contain the pandemic besides online learning. They provided guidelines to instruct people to fight against the pandemic and correct the misunderstandings of the pandemic [19]. They may provide constructive suggestions for people to keep social distance, take effective measures, and enhance public awareness of the pandemic to minimize the negative impact of the pandemic through social media [20]. In Brazil, students changed their roles from learners to volunteers who distributed healthy information to the public and informed them of the right measures to contain the pandemic [21].

Students in online learning environments have played important roles as a cooperator, coordinator, communicator, leader, investigator, practitioner, effort maker, and responsibility assumer [22]. Students have been expected to provide cooperation with peers and teachers through online learning platforms, to coordinate learning tasks to maximize the effect of learning, to communicate with peers and teachers to express themselves and share opinions, to act as team leaders to complete learning tasks based on learning plans, to search for related information, solutions, and other necessary sources before posting them on learning platforms for peers to share and discuss, to do exercises, download, or upload learning resources independently, to design websites and make every effort to solve problems they meet, and to assume responsibility for the work they completed.

Students could act as the roles of both assessors and assessees in online learning. As assessors, students, similar to teachers in traditional learning, could review peers' assignment, homework, or online performance. As assessees, online learners, as with students in traditional learning, could improve their performance, assignment or homework based on feedback or comments from teachers [23]. In other words, online learners could act as both teachers and students. The former, acting as assessors, would evaluate online learners' performance and quality. The latter, acting as assessees, would keep pace with online learning progress and improve their online learning performance based on suggestions from teachers or peers.

During the lockdown period caused by the COVID-19 pandemic, student roles were reduced to conflicts with other competing roles such as babysitting and cooking. Thus, online learners were less motivated to complete assignments and less encouraged to focus on learning activities than traditional students. It is thus necessary for online course designers to find out strategies to decrease the role of conflicts and bolster learning roles. They may require students to concentrate on the learning activities and expose them to supervision with cameras on. Either regular or irregular reminders could also be sent to students to inform them of the incomplete assignment. Those who completed the assignment might be rewarded while those who did not might receive lower evaluation. In general, teachers and designers should weaken the erosion of student roles by minimizing the role conflicts.

### 4.2. RQ2: What Are the Changes in Digital Literacy during the COVID-19 Pandemic?

After the outbreak of COVID-19, learners' digital literacy significantly improved compared with that before the pandemic. The online learning during the pandemic time has forced learners to improve their digital literacy to participate in online academic activities and enhance their self-regulation [24]. The COVID-19 pandemic has increased the digital literacy by improving the contacts of learners with information technologies. The online learning environments have encouraged students to try various applications of online technologies, to receive information through online platforms, and to acquire knowledge through repetitive technological attempts. All of these have enhanced their digital literacy. The exterior representations of digital literacy could include the use of digital devices, the treatment of information, the communication through online platforms, technological ethics, and the information security.

The level of digital literacy varies from one learner to another despite the general improvement in digital literacy during the pandemic. Variables such as age, educational background, gender, educational institute, and preference could exert a great influence on the level of digital literacy [25]. When designing online courses during the pandemic time, teachers and developers should cater the courses to different characteristics of learners. They could provide online courses with fewer technical requirements for those with poorer educational backgrounds and those with a lower level of digital literacy. Compared with females, males tended to prefer technology-assisted learning styles. Designers and teachers could thus divide online courses into gender-specific styles. Younger generations tended to accept online learning more readily than older generations. Strongly self-regulated learners held more positive attitudes towards online learning than those with weaker self-regulation [26].

With the increasing use of digital technologies, digital literacy was significantly improved during the pandemic compared with that before the pandemic. The COVID-19 pandemic has greatly popularized the use of digital technologies either in or out of classrooms, which could gradually enhance the level of students' digital literacy. For instance, engineering students at the Vellore Institute of Technology in India valued the digital technologies and had a higher self-perceived digital literacy than before the pandemic. They opined that their digital literacy could be further enhanced to meet the challenges of the pandemic and that digital technologies could be used in English language education [27]. To develop advanced digital technologies is a good way to improve digital literacy and address the concerns with technological unfamiliarity.

### 4.3. RQ3: How to Improve Learning Achievements during the COVID-19 Pandemic?

Technological assistance is beneficial to online learning achievements. Although some studies have reported positive online learning achievements, most of the students have experienced online learning assisted with technology [28]. Those receiving online learning without any technological assistance tended to negatively evaluate online learning and also obtained poor learning achievements. In this way, students felt uncomfortable and unsatisfied with online learning despite their freedom to share opinions with peers and teachers. Advanced technologies could increase learning engagement and enrich academic activities [29].

Learning achievements could be greatly improved through digital technologies such as augmented reality (AR), virtual reality (VR), and computer games. AR-assisted learning could enhance students' learning interest, fortify their learning motivation, and improve their learning achievements. A video-based VR educational system could improve students' learning achievements, foster their learning attitudes, enhance their self-regulation, and strengthen their self-efficacy. The VR system could also positively influence learners' cognitive loads [30]. VR technologies could increase students' learning achievements in science courses. Computer games could improve online learners' self-efficacy, enhance their online learning engagement, and increase their satisfaction levels, followed by improved online learning achievements. Computer games could address this lockdown by providing instructive strategies in the online context. As an innovative teaching approach, online computer game-assisted pedagogy could improve educational outcomes and thus gain popularity [31]. Featuring joyfulness and voluntariness, online computer games could improve online learning effectiveness by increasing learning engagement and enriching learning activities.

Information technologies could be used to improve students' online learning achievements. For instance, an online project-based learning platform could provide nearly the same high-quality online learning and teaching as the face-to-face method. Other information technologies such as cloud computing, Rain Classroom [32,33], and massive open online courses [34] could also improve the online learning quality. Based on information technologies, online learning platforms could enable teachers to supervise students' learning progress and learning behaviors such as attendance, time span, discussion, interactions, question answers, and completion of assignment. The learning achievements might then be enhanced due to teachers' supervision. Students' interactions with peers and teachers also revealed their learning performance and thus enhanced their learning achievements.

A social regulation-based learning style could improve online learning achievements during the pandemic. Social regulation indicates the combination of self-regulation and communication of learning strategies. Researchers have attached great importance to social regulation in collaborative learning. The strategy of social regulation could help students regulate their learning behaviors and improve online learning achievements [35]. A social regulation-based learning strategy could greatly promote online learning achievements by enhancing learning motivation. The social regulation strategy outweighed the traditional strategy where students merely focused on examination scores and teachers' guidelines. The former can connect students with peers and teachers through interactions, which can influence their learning behaviors and enhance their self-regulation in online learning environments.

A combination of interactions and self-regulation can increase online learning achievements. Strong self-regulation can encourage students to complete learning tasks on schedule and thus facilitate their learning effectiveness and efficiency and improve their online learning achievements. Interactions with peers or teachers can complement the lack of knowledge and promote students' mutual perceptions towards knowledge. The process of knowledge acquisition and interpersonal relationship can be maintained and continued through interactions and self-regulation. Similarly, it has been concluded that social regulation can exert a positive influence on learning performances [36]. Regular or irregular online examinations with proper supervision can also enhance learners' self-regulation and peer interactions. Students may interact with peers or teachers with a view to obtaining higher scores in online examinations. Mutually interactive learning strategies and goals can encourage students to participate in learning activities and thus boost their learning achievements.

The improved social and cognitive engagement can improve students' online learning achievements. In online learning environments, students were expected to be socially and cognitively engaged in learning activities such as knowledge acquisition and information retrieval. In face-to-face classroom-based learning, students can conveniently engage in learning activities under the supervision of teachers and be motivated by peers. In online learning environments, teachers can also track students' learning activities through the online platform, which encourages students to participate in activities and complete assignments on schedule. Students are expected to pay enough attention to the online teaching and acquire the knowledge delivered online in time. Otherwise, they will fail to keep pace with peers and teachers. Proper cognitive loads can be maintained through a reasonable amount of social engagement in online learning activities, which can improve online learning quality. All of these strategies might improve students' learning achievements in online learning environments.

However, inconsistent findings have been revealed regarding the students' online learning achievements. Success in online learning environments and continuance of online learning styles can be achieved by the students with positive perceptions towards online learning, mutual interactions, timely feedback from teachers, and strong self-organization abilities. University lecturers and educational management have been unsatisfied with online learning. They have found several disadvantages to the online medium such as paucity of enthusiasm in online education, lack of credibility of online certificates, poor digital infrastructure, technical issues, paucity of practice in online education, lower levels of interaction, poor mastery of online knowledge, and lack of interest in online education. University lecturers and preservice teachers should receive online pedagogical training and technological enhancement. Participants have also noted the need for pedagogical and technological training during the lockdown period. Nevertheless, preservice teachers have positively evaluated online learning and also obtained favorable online learning achievements. English language learners with strong self-efficacy also achieved success in online language learning.

### 4.4. How to Improve Online Learning Motivation during the COVID-19 Pandemic?

Students' online learning motivation could be enhanced by teachers through various strategies. The timely feedback from teachers could motivate students to engage in learning activities [37]. It was hard for the teachers to identify students' learning behaviors and progress since they could seldom supervise the whole learning scenario through the online learning platforms. They were unable to see whether students were able to understand the knowledge delivered online. They did not exactly master students' learning progress and thus tended to teach without the knowledge of students' perceptions. Interactive strategies such as quizzes, question answers, and progress checking could help teachers to perceive students' learning pace and modulate their teaching progress. Reminders from teachers could motivate students to complete learning tasks and assignments on schedule. Those who fail to complete tasks should receive warnings or negative evaluations. Teachers could motivate students to engage in learning by reminding them through social media, email, phone, text, or online interactive platforms.

Cooperation between teachers and learners could enhance online learning motivation. Teachers and designers could enhance learning motivation via effective online pedagogical approaches, online learning platforms, online curricular contents, and teaching plans. In the online environment, scaffolding teaching was necessary for teachers to adapt to the new teaching environment. Students should be motivated to cooperate with online teaching where self-regulation and motivation could predict their online learning success. Teachers could take effective measures to enhance students' motivation, e.g., badges, punishment, timely feedback, and reminders. Learning motivation has greatly influenced learner engagement in learning activities, especially during the COVID-19 pandemic. Students should understand the special learning environments, increase their engagement, and regulate their other learning behaviors. They should also foster their self-regulation, self-efficacy, and motivation actively in online learning environments.

Gamification or computer applications could also improve online learning motivation. Students and their families positively evaluated gamified computer applications. Gamification could improve learners' critical thinking abilities, achieve their success in learning, enhance their participation in learning activities, fortify their interactions, and foster their online learning motivation [38]. Gamified computer applications could enhance their motivation of mathematics learning [39]. Augmented reality applications could enhance learning motivation and concretely present the abstract knowledge. Designers and developers could integrate gamified elements into the computer applications to enhance online learning motivation. They could also make the online platforms more attractive and amusing by including gamification. Perceived usefulness and perceived ease of computer applications are also important predictors of students' acceptance. Developers could make them easier and more useful to online learners [40]. We summarize these findings in Figure 4.

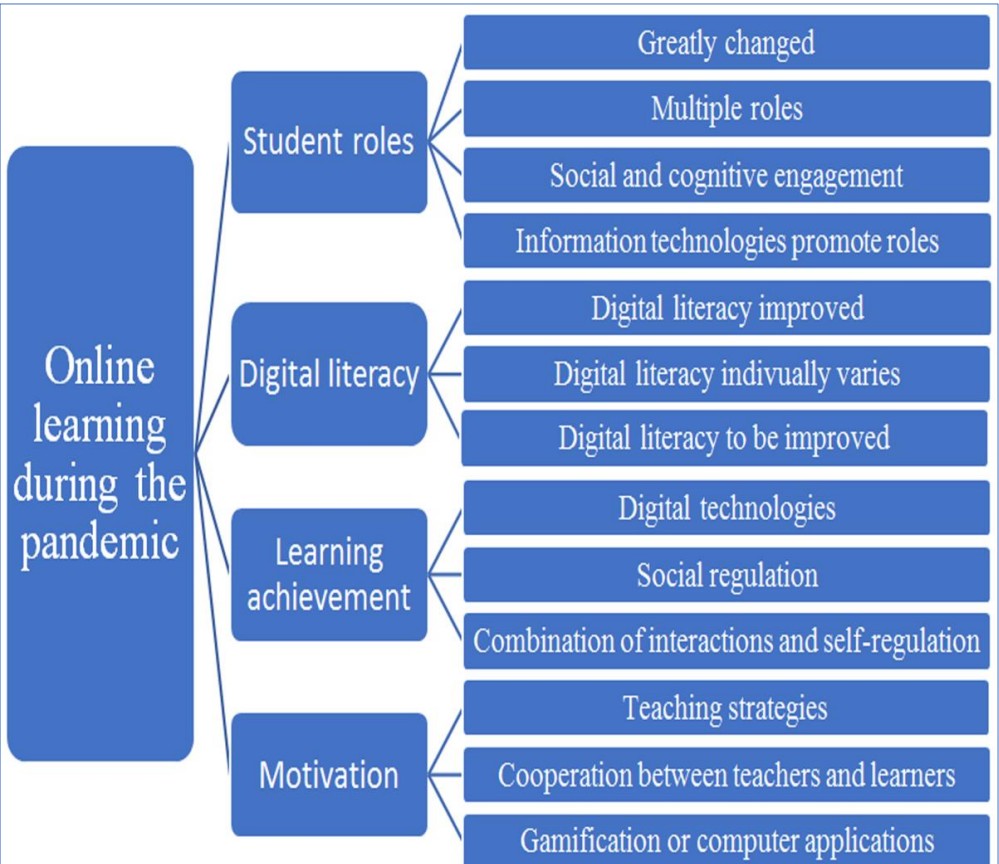

**Figure 4.** Sustaining student roles, learning achievements, digital literacy, and motivation.

## 5. Discussion

### 5.1. Vagueness of Student Roles and Their Ethics

Although numerous studies have identified different student roles in online learning environments during the pandemic, they still remain unclear and even difficult to be determined. COVID-19 has raised the question regarding the correct student role under an emergency condition, especially for those medical students who acted as complicated roles. Other major figures should recognize the seriousness of the pandemic and deliver the knowledge to the public. They should also cooperate with teachers and designers and accept online learning styles with strong self-regulation. To support students as online learners or other unknown roles, seamless learning should be well prepared and designed including both online and offline learning. The deficiencies of systematic curricular design were exposed to the public when it was confronted with the pandemic, which has made student role vaguer and more complicated.

During the pandemic, ethical issues in online learning became a serious concern [41]. Personal information is vulnerable in online learning environments. Students should make sure that their personal information is protected and should meanwhile protect others' when receiving online education. The sudden restrictions and lockdown caused by the pandemic could lead students to feel that they are suddenly beyond any support. They may also possibly feel it hard to strike a balance between their roles as students or family members [42]. In online learning environments, they tended to be required to participate in teamwork and cooperative activities, where their student role was diminished [43]. In teamwork activities, students should make every effort to engage in academic activities, be helpful to others, and be aware of the privacy protection.

### 5.2. Paying Close Attention to Digital Literacy in Online Learning

Digital literacy may exert a great impact on online learning achievements [44]. Students with stronger digital competencies have tended to be more motivated and interested in online or blended learning, coupled with more self-regulation and engagement in learning, especially during the pandemic. The perceived digital literacy could also improve students' learning motivation to achieve online learning goals. Those with poorer digital literacy have been subject to technological problems and have felt it difficult to adapt to the online learning style [45]. Whether for a younger or older generation, digital literacy plays an important role in education during or after the pandemic.

Digital literacy can greatly facilitate the online learning and teaching process. Digital literacy could help learners and teachers address technical issues and online learning risks such as privacy breaches [46]. Digital literacy has been an essential component in facilitating the use of educational technologies, academic communication, learning activity organization, learning performance assessment, and information distribution. Digital literacy and parental assistance could help school students address online learning challenges and improve online learning and teaching process during the pandemic [47]. Teachers with higher levels of digital literacy could deal with an emergency situation in a proper manner and with easy access to online learning resources. They, considering the pandemic a new opportunity, have been ready to cater their teaching styles to the new requirements during the pandemic [48].

During the lockdown time, it is necessary to improve digital literacy of both teachers and students. Higher levels of digital literacy could improve online interactions and new teaching styles [49]. Digital literacy could directly influence learning achievements [50]. With proper digital literacy, students could obtain a large amount of learning information from the online learning platforms and maintain interpersonal relationships with peers and teachers. However, educational institutes and educators have not been well prepared for the pandemic where higher levels of digital literacy were needed [51]. Educators could improve students' levels of digital literacy by developing courses specialized in computer technologies and the use of digital tools in the form of offline, online, or blended methods.

### 5.3. Assessment of Online Learning Achievements

It is not easy to assess online learning achievements fairly and appropriately in an online learning environment despite its importance in online learning. The online assessment paradigms of learning achievements are quite different from traditional methods. Since the abrupt shift to online learning, various formative assessment paradigms have been adopted, e.g., online quizzes or exams, online engagement (responses, feedback, video watching period, or online discussion), peer interactions, and peer assessment. Most of the assessment paradigms are realized through learning management systems [52].

Online examinations may act as a tool to facilitate online learning achievements. The summative assessment paradigm, e.g., online final exams, is still useful to online assessment although it is difficult to be implemented due to different levels of digital literacy, cheating, technical problems, poor internet connection, and underdeveloped digital infrastructures. Online examinations, different from traditional examinations, may be implemented on learning platforms, where students' scores can be easily collected and analyzed. Teachers can master the learning performances easily and conveniently so that they can design the teaching progress. Therefore, it is worth paying attention to final credits and exams for courses or subjects implemented with the use of e-learning technology. It is thus an important part of online learning assessment.

### 5.4. Paying Close Attention to Motivation in Online Learning

Motivation can exert an important influence on online learning and should thus have the appropriate level of importance attached to it. The sudden shift to online learning greatly influenced learning motivation, which could tremendously influence online learning achievements [53], learner satisfaction [54], and learner participation in online learning environments. The major concern in online learning environments is whether students are motivated or compelled to learn. Teachers tend to be ignorant of this. They are expected to be familiar with students' needs and their potential conception by obtaining feedback from the online learning platforms in a timely manner. Learning motivation has been a variable subject to the specific learning environment and learner personalities.

## 6. Conclusions

### 6.1. Major Findings

Through a rapid evidence assessment review study based on the protocol of Preferred Reporting Items for Systematic Review and Meta-analysis (PRISMA), it is concluded that student roles have experienced great changes, that students have assumed multiple roles, that they are socially and cognitively engaged in their roles and that these roles can be promoted by information technologies. Their digital literacy, individually varying, needs to be improved, although it has undergone improvements. Digital technologies and social regulation can improve online learning achievements. Teaching strategies, teacher–student cooperation, gamification, and computer applications can improve online learning achievements.

### 6.2. Limitations

There are still some limitations to this study although it has been conducted based on the framework of PRISMA. Firstly, the researchers could not obtain all the high-quality literature due to the limitation of library resources. Secondly, this study conducted a rapid evidence assessment review method without statistical support. Thirdly, this study could not include all the factors in online learning environments during the pandemic. There may be some other influencing factors beyond this study.

### 6.3. Future Research Implications

Future research could focus on inter-disciplinary research into the models to sustain online learning during or after the pandemic. Researchers could develop serious games or computer applications to improve online learning effectiveness during or after the

pandemic since gamification and computer applications could improve online learning achievements [55]. The theory of social regulation [56] could also be further explored and applied to online learning environments.

Online examinations can facilitate online learning through their advantages. The COVID-19 pandemic has changed traditional pencil-based examinations to online examinations [57]. Online examinations can shift traditional grades and assignment to peer discussion, opinion sharing, and online interactions to reduce students' anxiety and enhance their critical thinking abilities, as well as collaborative learning skills. Online examinations allow students to participate in learning at their own convenience, which can relax students and cultivate comfortable learning environments [58]. The learning management system can record the completion time for specific examinations and collect data much faster than the traditional format. Teachers can review individual learning outcomes conveniently.

Online examinations may be considered important in future research into online learning. In view of the popularity of online examinations, future research can shed light on how to design and improve online examinations and how to address challenges in online examinations. Teachers and designers can examine the approaches to online examinations, and to the credibility of checking the knowledge and skills of students without close supervision. Artificial intelligence technologies can be integrated into online examinations to improve online learning effectiveness and efficiency. It is also indispensable to conduct survey research with the participation of teachers and students to effectively assess online or distance learning outcomes in the future.

**Funding:** This research was funded by 2019 MOOC of Beijing Language and Culture University (MOOC201902) (Important) "Introduction to Linguistics"; "Introduction to Linguistics" of online and offline mixed courses in Beijing Language and Culture University in 2020; Special fund of Beijing Co-construction Project-Research and reform of the "Undergraduate Teaching Reform and Innovation Project" of Beijing higher education in 2020-innovative "multilingual +" excellent talent training system (202010032003); The research project of Graduate Students of Beijing Language and Culture University "Xi Jinping: The Governance of China" (SJTS202108).

**Institutional Review Board Statement:** Not applicable.

**Informed Consent Statement:** Not applicable.

**Data Availability Statement:** Not applicable.

**Acknowledgments:** We would like to extend our gratitude to anonymous reviewers and funds that financially supported this research.

**Conflicts of Interest:** The author declare no conflict of interest.

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
