# Peer review of "Sustaining Student Roles, Digital Literacy, Learning Achievements, and Motivation in Online Learning Environments during the COVID-19 Pandemic"

_sustainability, doi:10.3390/su14084388_

Round 1

Reviewer 1 Report

In 3.1, the authors make the following modifications, "Different from traditional face-to-face learning, students in online learning environments played important roles as a cooperator, coordinator, communicator, leader, investigator, practitioner, effort maker, and responsibility assumer[22]." However, such student roles were already introduced in face-to-face learning in the form of group work and PBL with the introduction of Active Learning. So, it is not appropriate to describe this as a change from face-to-face education. Similarly, the authors state that "Different from traditional face-to-face learning, students could act as the roles of both assessors and assessees in online learning." in revised section 3.1. Such Peer Review is also possible in face to face learning, and it seems strange to address it as a change due to online learning (although there may be cases where online learning facilitates it).

Author Response

Dear Reviewer,

Thank you very much for your time spent reviewing our study. We have carefully revised it based on your constructive comments. Please see the attachment for details. We would greatly appreciate you if you could seriously consider our revisions.

Best wishes!

Sincerely yours,

Author

Reviewer 2 Report

Why the set of analyzed databases included databases in the field of chemical sciences, i.e. Current Chemical Reactions, Index Chemicus (IC). As I noticed, the primary research bases were those in the humanities and sociological sciences. What was unique about the databases in the field of chemical sciences, if they were included in the research on educating students in the pandemic period?

Most databases were analyzed, as shown in the paragraph on lines 112-124, from the 1980s / 1990s to 2022. The Science Citation Index Expanded (SCI-EXPANDED) database, however, as shown in line 117, was analyzed since 1900. This is a surprising date for me, which is completely inconsistent with the dates of searches that were given for other analyzed databases. Is 1900 a mistake?

I don't understand why the figures in Figure 1 are presented in the order from 2022 to 1999. The trend shows itself in subsequent years rather than in the years going backwards. In my opinion, you have to reverse the order of the years in Figure 1. Moreover, on the axis of ordinate (y) in Figure 1 you need to write what the data refer to. My guess is that this is the number of publications found in the databases, but this information must be included in the description of the ordinates.

In the analysis presented in the chapters Results and Discussion, it would be worth paying attention to final credits and exams for courses / subjects implemented with the use of e-learning technology. I think this is an important part of the distance learning assessment. Based on the analysis of publications, it would be worth indicating the approach of teachers and students to conducting examinations, to the credibility of checking the knowledge and skills of students. Are there any trends in this area, or maybe some suggestions regarding the methods of conducting examinations in student groups? It is worth developing this topic of research and discussion.

I think that in the Conclusions chapter, in the Future research implications section, it would be worth mentioning the need to conduct survey research with the participation of teachers and students. Based on the results of his own survey, the author could verify the issues raised in the article related to the assessment of distance learning by students.

In References, publications 25 and 26 are exactly the same publications. One of them has to be removed and renumbered in the text. The same remark applies to publications 30 and 32 in References. These are also the same publications and one needs to be deleted. The same remark applies to publications 31 and 35 in References. These are also the same publications and one needs to be deleted. The same remark applies to publications at numbers 6 and 48 in References. This is exactly the same publication and should not be listed in References twice. In general, I suggest the Author carefully review the publications in References and remove those publications that repeat the publications already cited.

Author Response

(The authors gave the same response as above.)

Round 2

Reviewer 1 Report

The modified version can be accepted.

This manuscript is a resubmission of an earlier submission. The following is a list of the peer review reports and author responses from that submission.

Round 1

Reviewer 1 Report

This article is a literature survey on online learning environments under COVID-19. The RQs were clearly defined, and the target literature was referenced appropriately.

However, the findings are not always reasonable to the RQs. RQ1 is about the role change of students during COVID-19. Although sections 3.1 and 4.1 provide examples of medical students or others, the authors do not directly answer this RQ (What are changes in student roles ...). It is necessary to comprehensively discuss what roles usual students have shifted to and what learning activities and communication are now required. In addition, the logical structure was unclear since some of the discussions were strongly related to the learning achievements of RQ3, such as the last two paragraphs in section 3.1.
Similarly, RQ2 describes the changes in digital literacy at the COVID-19. However, there is no discussion of how the digital literacy of teachers and students changed before and after COVID-19. These gaps between the RQs and the findings make it difficult to understand the contribution of this article.

In addition, RQ3 discusses learning achievements. However, it is not easy to assess them fairly and appropriately (especially summative assessment) in an online learning environment. It is necessary to discuss the changing paradigm of assessment of learning achievements and the usefulness of this discussion in achieving new contributions. Moreover, there is much overlap between chapter 3 and chapter 4. It makes the findings of this survey difficult to understand for the reader.

The title of the article should also emphasize that it is a literature survey to make the target of this research clearer.